# A hybrid deep learning and residual connection-based architecture for intrusion detection in autonomous vehicles

Hareem Kibriya[1], Ayesha Siddiqa[2], Saad Alahmari ID[3], Wazir Zada Khan ID[2]*,
Saad Nasser Altamimi[4], Atta ur Rehman Khan[5]

1 Department of Computer Science, Air University, Islamabad, Pakistan, 2 Department of Computer Science, University of Wah, Wah Cantt, Pakistan, 3 Department of Computer Science, Applied College, Northern Border University, Arar, Saudi Arabia, 4 Department of Information Systems, College of Computer and Information Sciences, Imam Mohammad Ibn Saud Islamic University (IMSIU), Riyadh, Saudi Arabia, 5 College of Engineering and IT, Ajman University, UAE

* wazir.zada.khan@uow.edu.pk

## Abstract

The emergence of Autonomous and Connected Autonomous Vehicles (CAVs) has transformed the automotive landscape drastically over the past few years by offering enhanced features in the vehicles for drivers' safety and convenience. These developments have introduced various features in AVs i.e., lane-keeping, cruise control, etc. These features are mainly powered by the Electronic Control Units (ECUs) that communicate using the Controller Area Network (CAN) bus protocol. The components in the AVs communicate with each other by sending and receiving messages via the CAN bus. However, despite increased connectivity, these vehicles have become vulnerable to cyber attacks, as malicious actors can exploit the CAN protocol to manipulate vehicle behavior, which can not only threaten the safety of the passengers but public as well. Hence, several Intrusion Detection Systems (IDS) have been proposed, however, these systems struggle with computational complexity, limited effectiveness against sophisticated attack types, and a lack of interpretability and transparency of detection mechanisms. To address challenges in the existing systems, this paper presents a novel hybrid Deep Learning (DL)-based IDS using DL components such as Convolutional layer and Long Short-Term Memory (LSTM) layers to capture complex patterns in the CAN messages. The proposed IDS uses a residual connection to enhance gradient flow and training stability. The system is evaluated on four common attack types, namely RPM Spoofing, Gear Spoofing, Fuzzy, and Denial of Service (DoS), achieving a detection accuracy of 99.99%. Finally, the outcomes of the proposed IDS are visually interpreted using the Explainable AI (XAI) technique called Local Interpretable Model-agnostic Explanations (LIME) to provide transparency into the model's decision-making process, thus increasing trust in the system's deployment in real-world AV environments.

**Data availability statement:** This study exclusively uses a publicly available dataset, which is fully described in the manuscript and accessible through the referenced public repository (https://ocslab.hksecurity.net/Datasets/car-hacking-dataset).

**Funding:** This study was financially supported by the Deanship of Scientific Research at Northern Border University, Arar, KSA in the form of a grant provided to SA (NBU- FFR-2026-451-02).

**Competing interests:** The authors have declared that no competing interests exist.

## Introduction

Still now passenger and driver safety remains a pressing global issue. In 2021 alone, almost 1.19 million people succumbed to road accidents worldwide, meaning almost 15 deaths per 0.1 million people. In 2019, road traffic injuries were ranked as the 12th leading cause of death globally, also a main cause of death among children and young adults aged between 5–29. According to a report by the WHO, by the time one finishes reading a single page of their report, at least five people will have lost their lives in road traffic crashes. These figures highlight the critical need for advanced and responsive safety systems to reduce the risk of traffic-related fatalities [1].

The automotive industry is going through a major change with the growth of autonomous vehicles (AVs). They offer workable ways to address long-standing concerns related to road safety and traffic flow. AVs rely on advanced sensing units and real-time data processing to read and respond to changes in the driving environment more consistently than human drivers. This change has lowered the effect of human error, still the leading cause of road accidents. Reliable AV operation depends on stable intra-vehicle communication systems that collect and share data from several sensors and On-Board Units (OBUs) [2]. These data streams support timely decisions and coordinated control actions [3,4].

The CAN bus protocol manages critical communication between components of AVs by facilitating real-time data exchange among ECUs, which are responsible for essential vehicular functions such as braking, acceleration, and steering [5–7]. However, the CAN protocol lacks basic security features to detect forged packets injected with a malicious intention to manipulate the vehicle, endanger the passengers, and also surrounding road users [8,9].

Hence, to overcome this limitation, numerous IDS have been proposed to enhance CAN bus security. These systems include rule-based, ML-based, and DL-based methods. Traditional rule-based IDSs are limited to manual feature engineering and rule definition, thus limiting their adaptability and scalability in a scenario of evolving attack patterns. DL based models are hence being developed to overcome limitations in the traditional IDS. These systems automate the process of feature engineering, learning, and classification directly from the data without predefined rules or human intervention. Hence, these systems are scalable to ever-changing attack scenarios [10]. Despite considerable advancements, these systems still face several critical limitations. One major challenge with the computational complexity of these algorithms is the high to be deploying them in AVs, as CAN protocol is designed for lightweight, real-time communication within resource-constrained environments. Hence, deployment of such a resource-demanding IDS may monopolize computational resources, therefore, may compromise the performance of other time-critical vehicle subsystems by resulting in delayed responses. Therefore, such memory-intensive models can impose significant pressure on the limited memory resources, thereby affecting the vehicle's overall system reliability [11,12].

Many AI-based intrusion detection systems (IDSs) lack interpretability, another significant problem. Deep learning models, in particular, are sometimes referred to as black boxes since their fundamental logic is concealed. When the reasoning

underlying a model's output cannot be articulated, it gives rise to trust issues. These problems are significant, especially in autonomous driving, where every choice can have real-world consequences [7,13,14]. In order to address these deficiencies, we propose a lightweight explainable hybrid deep learning model that employs a residual connection to detect four types of cyber attacks in AV systems. The main contributions of our work are outlined below:

- We propose an intrusion detection system for CAV that combines convolutional layers with hybrid LSTM–GRU units and residual connections. The proposed hybrid design captures both temporal and spatial patterns in CAN bus traffic while keeping the parameter count to around ≈ *24K*.

- Due to its lightweight design and low computational overhead, the proposed model is suitable for resource resource-constrained environment

- We utilized a multi-seed data splitting approach to ensure the proposed model's reliability

- We perform extensive ablation studies to find the best architectural setup and hyperparameter settings

- Local Interpretable Model-Agnostic Explanations (LIME) visualization is incorporated

The rest of the paper is structured as follows: The Related Work section reviews existing research. The Proposed Method section provides details of the proposed IDS system. The Results and Discussion section covers experimental setup, evaluation metrics, multi-seed data splitting analysis, ablation results, and XAI visualizations. Finally, we conclude the study.

## Related Work

Intrusion detection for CAN communication has evolved significantly over the last decade. Research ranges from traditional threshold-based methods to deep learning, federated learning, and explainable AI frameworks. In this section, we grouped prior work into four categories: (A) traditional and statistical IDS techniques for CAN bus security, (B) deep and transfer learning-based IDS models, (C) federated and collaborative intrusion detection frameworks, and (D) explainable and emerging intelligent IDS systems. Table 1 summarizes representative methods. While classical methods offer transparency and efficiency, deep and collaborative learning models provide stronger adaptability but often at the cost of interpretability, an issue that motivates our lightweight, explainable IDS approach.

### A. Traditional and Statistical IDS

A threshold-based anomaly detection scheme is proposed by Lee et al. [15]. They introduced an Offset ratio and Time interval-based Intrusion Detection System (OTIDS). The offset ratio and time intervals between request and response messages were analyzed. Their approach effectively detected message injection and impersonation node attacks by identifying deviations from normal response patterns. OTIDS maintained a high instant reply ratio (around 100%) under normal conditions, which dropped significantly during attacks, such as DoS, to about 10–20%. The lost reply ratio increased from less than 0.1% in attack-free states to over 1% during attacks. The correlation coefficient between offsets and time intervals was near 1 in normal conditions but dropped significantly during an attack to indicate anomalies [16]. The authors implemented their prototype with a Raspberry Pi3. Lee et al. [15] released their dataset named OTIDS for intrusion detection in CAN with four attack types. Their dataset is publicly available on the website of the Hacking and Countermeasures Research Lab (HCRL).

SIDuDTW framework for detecting malicious messages in CAN bus traffic uses Dynamic Time Warping (DTW) to compare CAN ID sequences [17]. SIDuDTW focuses on finding recurring patterns in the data. It also uses wave splitting techniques to improve detection efficiency. It was tested using publicly available Car Hacking datasets and real vehicle data. On the Car Hacking datasets, SIDuDTW achieved a detection accuracy of 96.67%, a G-mean of 97.15%, and an

**Table 1. Overview of intrusion detection techniques in CAN.**

| Article | Approach | Key Features | Dataset | F1-Score | Attacks Detected | Limitations | XAI |
|---------|----------|--------------|---------|----------|-----------------|-------------|-----|
| Lee et al. [15] (2017) | OTIDS | Threshold-based detection analyzing CAN message offsets and time intervals | OTIDS | – | DoS, Fuzzy, Impersonation | Relies on remote frame and requires additional hardware | × |
| Tariq et al. [18] (2020) | CAN-Transfer | ConvLSTM with Transfer Learning for spatial-temporal pattern analysis | KIA Soul and Hyundai Sonata datasets | 95.25% (known), 88.47% (unknown) | Replay, Injection, Spoofing, DoS | Impressive performance on unknown attacks but needs validation in production | × |
| Amato et al. [7] (2021) | NN and MLP | Uses CAN message data bytes for feature extraction | Car Hacking Datasets | 100%(RPM), 98.2%(DoS), 100%(Gear), 99.6%(Fuzzy) | DoS, Fuzzy, Gear Spoofing, RPM Spoofing | Performance decreased with deeper networks | × |
| Javed et al. [19] (2021) | CANintelli-IDS | CNN+GRU with attention mechanism | OTIDS | 94% (single), 93.79% (mixed) | Injection, Replay, Spoofing, DoS | Needs improvement for unknown attacks | × |
| Wang et al. [20] (2021) | PDT and EDC | LSTM-based deep learning models for time-series prediction and reconstruction | Simulated datasets | 67%–91% (varies by attack) | Stealthy, Replay, Injection | Struggles with unknown attacks | × |
| Sun et al. [17] (2022) | SIDuDTW | DTW for sequence similarity in CAN ID messages | Car Hacking Datasets | 94.88% | DoS, Fuzzy, Injection | False positives from CAN noise | × |
| Yu et al. [12] (2022) | Federated Learning with LSTM | LSTM predicting CAN message IDs and detecting anomalies | HCRL (train), synthetic (test) | 89.4%(Replay), 90%(Spoofing), 92.2%(Drop), 96.9%(DoS) | Spoofing, Replay, Drop, DoS | Weak generalization across datasets | × |
| Wang et al. [11] (2023) | TLSIDS | DenseNet+GAN-based classification with self-learning | Car Hacking Dataset | 99.93% | DoS, Gear Spoofing, RPM Spoofing | High computational complexity | × |
| Hoang et al. [22] (2023) | CANPerFL | Federated Learning across vehicles | Real CAN data (KIA, BMW, Tesla) | 90%+ (with 30k samples/model) | Spoofing, Replay, DoS, Injection | Generalization affected by car model differences | × |
| Wickramasinghe et al. [23] (2023) | RX-ADS | ResNet Autoencoder+adversarial training | OTIDS and Car Hacking Datasets | 99.67%(DoS), 99.51%(Fuzzy) | DoS, Fuzzy, Spoofing | Limited adaptability to unseen threats | ✓ |
| Anbalagan et al. [24] (2023) | IIDS | CNN with hyperparameter tuning for AV security | Simulated datasets (NS2, SUMO) | 98% | Injection, Spoofing, DoS | Simulated data lacks real-world complexity | × |
| Alsulami et al. [21] (2023) | Transfer Learning for AV-CPS | CNN on sensor data converted into images | Simulated AV-CPS dataset | 99.47% (GoogLeNet) | Injection, Replay, DoS | Simulated only; needs real-world testing | × |

F1-Score of 94.88%. These results were consistent across five different types of attacks in real vehicle environments. SIDuDTW could not eliminate false positives due to the inherent noise in CAN bus traffic and due to the occasional misclassification of normal waves as attacks. It is designed to detect malicious messages in wave sequences.

## B. Deep and Transfer Learning-based IDS

In this subsection, we reviewed recent deep learning-based frameworks that are capable of capturing the nonlinear temporal-spatial dependencies in CAN traffic. For instance, CANTransfer [18] employs a deep learning approach with transfer learning to detect both known and new types of attacks. It analyzes the spatial and temporal patterns in CAN traffic. CANTransfer addresses the challenge of detecting unknown attacks with limited data by employing one-shot learning to adapt to new intrusion types. It was evaluated using datasets from the KIA Soul and Hyundai Sonata. It outperformed

baseline models as it obtained an F1-score of 95.25% for known attacks and 88.47% for unknown attacks. CANTransfer demonstrated sufficient processing speed for real-time operation. It maintained low loss and improved accuracy across multiple trials. Its improved performance was due to transfer learning, which enables it to be more robust and adaptive intrusion detection for CAN systems compared to OTIDS.

Amato et al. [7] utilized neural networks and Multi-Layer Perceptrons (MLPs) to classify CAN messages. The features used for classification are derived from the data bytes of CAN packets. It involves creating a feature vector from these data bytes to distinguish between normal and malicious messages. The method is validated using four real-world datasets, with attacks including DoS, Fuzzy attacks, Gear spoofing, and RPM spoofing. MLP models with 1 and 3 hidden layers performed best, and they achieved the highest precision up to 97.4% and recall up to 96.5% in detecting attacks. It demonstrated that the MLP model is effective in detecting four types of CAN bus attacks. The approach is tested against a limited number of attacks.

CANintelliIDS, as proposed by Javed et al. [19], integrates the advantages of feature extraction via Convolutional Neural Networks (CNN) and sequence learning through Gated Recurrent Units (GRU) enhanced by an attention mechanism. It possesses the capability to identify individual intrusion types (e.g., Denial of Service, Fuzzy, and Impersonation attacks) as well as composite intrusion scenarios wherein multiple attack types are perpetrated concurrently. The performance of CANintelliIDS was evaluated using balanced and imbalanced datasets taken from real-world car CAN bus data with a wide range of attack scenarios. The system obtained an average F1-Score of almost 94% for individual attacks and 93.79% for combined attacks. Its effectiveness could be increased by enhancing its ability to detect new, previously unknown attacks. On the other hand, Wang et al. [20] detected stealthy attacks on AV systems including Adaptive Cruise Control and Cooperative Adaptive Cruise Control. They created three types of stealthy attacks. These attacks could fool standard rule-based detection methods. The attacks threatened important system functions like collision avoidance and vehicle-following distance. To address these threats, the authors developed two deep learning models. The first is the Predictor-based model (PDT). It uses LSTM networks to predict normal behavior from time-series data. The second is the Encoder-Decoder model (EDC). It uses an LSTM-based encoder-decoder to reconstruct time-series data. The PDT model reached 97% accuracy when no attacks were present. It also showed strong detection performance, with F1-Scores between 67% and 91% across different attacks. Both models were trained without attacker-specific data. It could affect their performance on new attacks not seen during training. Alsulami et al. [21] explored transfer learning for detecting cyberattacks in autonomous vehicle cyber-physical systems (AV-CPS). They implemented the CAN communication protocol using Simulink's Vehicle Network Toolbox. This protocol was integrated into an AV simulation model. They converted sensor data from the simulation into images. These images represented both normal and attack scenarios. Eight pre-trained CNNs were trained on this image dataset. Among them, GoogLeNet gave the best results. It achieved an F1 score of 99. 47%. While Hassan et al. [3] presented a hybrid intrusion detection system for autonomous vehicles. The system combines Recursive Feature Elimination (RFE) for selecting features with Principal Component Analysis (PCA) for reducing dimensionality. They used a Deep Neural Network (DNN) for classification. They attained an F1-score of 98.5% on a publicly available Car-Hacking dataset. They did not integrate any XAI techniques, which could limit user trust in their model in real-world deployments.

## C. Federated and Collaborative IDS Frameworks

To address privacy and generalization challenges, several studies have employed federated and collaborative learning paradigms. For instance, Yu et al. [12] introduced a federated learning-based approach for intrusion detection. Their method exploits the periodicity of CAN message IDs to build a predictive model. The prediction of message IDs is used as a basis for network intrusion detection by recognizing deviations from expected patterns. Abnormalities in network messages are detected by comparing predicted message IDs against actual received IDs. The LSTM model is trained on an attack-free dataset of CAN messages from HCRL. For testing, the dataset was generated with abnormal values to

mimic four types of attacks, i.e., Spoofing, replay, drop, and DoS attacks. F1-Score of 90%, 89.4%, 92.2%, and 96.9% is achieved in case of spoofing, replay, drop, and DoS attacks, respectively. The federated-LSTM-based intrusion detection method achieved over 90% accuracy in identifying four types of malicious network attacks. The simulation results are based on a specific dataset and generated attack models.

Hoang et al. [22] introduced CANPerFL, a framework designed to enhance the performance of in-vehicle IDS by sharing knowledge across multiple vehicles. CANPerFL employs federated learning to allow multiple vehicles to collaboratively train a global model while keeping their data local. Real CAN data was collected from three different car models: KIA, BMW, and Tesla. Data includes both normal operations and attack scenarios (fuzzy and replay attacks). The system performs well with at least 30k samples per car model with F1-Scores of above 90%. Wang et al. [11] proposed a Transfer Learning-based Self-Learning Intrusion Detection System (TLSIDS). It operates through a cascade detection approach comprising four modules. The basic detection module uses DenseNet to quickly filter known attacks. The advanced detection module employs a GAN-based model to identify unknown attacks hidden within normal traffic. The unknown attacks classification module classifies and tags these unknown attacks to generate new training data. The self learning module updates the model using transfer learning techniques to improve detection performance. It is tested on the Car Hacking dataset from HCRL, containing samples of DoS, Gear, RPM attacks, and normal traffic. Recall of 99.97%, precision of 99.90%, and F1-Score of 99.93% were reported.

### D. Explainability and Emerging Intelligent IDS Models

Explainability and cross-domain intelligence are important to enhance trust and interpretability in IDS for AVs.

Wickramasinghe et al. [23] presented a ResNet autoencoder-based eXplainable Anomaly Detection System (RX-ADS). It combined interpretability with adversarial machine learning techniques. The RX-ADS framework integrates three key components. A window-based feature extraction mechanism is used to analyze CAN frame data. Autoencoder-based architecture is utilized for anomaly detection. Explanations of detected anomalies are generated using an adversarial machine learning module. The system was tested on two benchmark datasets, OTIDS and Car Hacking from HCRL. It examined various time windows for feature extraction to get the best results in terms of anomaly detection. RX-ADS showed an F1-score of 99.67% and 99.51% in detecting DoS and Fuzzy attacks, respectively.

The Intelligent Intrusion Detection System (IIDS) proposed by Anbalagan et al. [24] addressed vulnerabilities within the Internet of Vehicles (IoV) by using a modified CNN enhanced with hyperparameter optimization. It detects and categorizes malicious AVs within a 5G Vehicle-to-Everything (V2X) environment. The evaluation of IIDS is carried out through simulations using Network Simulator-2 (NS2) and Simulation of Urban Mobility (SUMO). The F1 score, precision, and recall of 98% are reported for IIDS in the simulation results. IIDS's performance is evaluated using simulations in NS2 and SUMO. Complementary works such as SDNTruth [25] and MF2S-CID [26] extended explainable and adaptive detection mechanisms to vehicular networks. SDNTruth utilizes the centralized control of Software-Defined Networking (SDN) and entropy analysis for DDoS mitigation, while MF2S-CID provides a generalized and interpretable IDS with dynamic classifier selection for multiple attack types.

### Proposed Method

This paper proposes a lightweight, AI-based IDS designed to detect cyberattacks in AVs through analysis of CAN bus messages. As illustrated in Fig 1, the proposed IDS comprises five key stages: (1) Dataset acquisition and preprocessing (2) training of the deep learning model after extensive architectural modifications (3) prediction of potential attacks, and (4) visualization of model outputs using XAI techniques. The system is trained and evaluated using a publicly available, benchmark Car Hacking dataset provided by the Hacking and Countermeasure Research Lab (HCRL), which includes four major attack categories, i.e., Denial of Service (DoS), Fuzzy, RPM Spoofing, and Gear Spoofing—alongside one normal class.

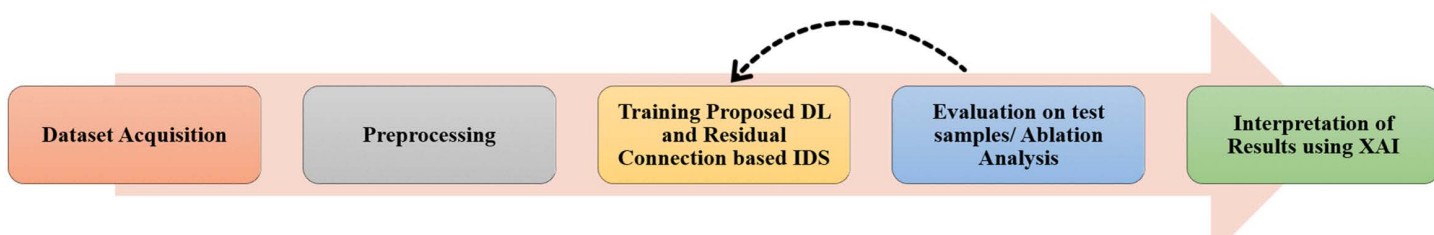

**Fig 1. Workflow Diagram of the Proposed IDS.**

Algorithm 1 presents an overview of the steps performed in this study.

**Algorithm 1 Proposed IDS for cyber attack detection on CAN bus with explainability**

```
1: Input: Set of CAN bus attack datasets DB = {DB₁, DB₂, DB₃, DB₄}
2: Output: Prediction P for each dataset and LIME-based explanations VR
3: For each DBᵢ in DB do
4:     DB_Dec ← HexToDecimal(DBᵢ)
5:     DB_Clean ← ReplaceNaN(DB_Dec, 0)
6:     D_DF ← CombineFeatures(DB_Clean)
7:     Pᵢ ← DeepResCAN(D_DF)
8:     eStore predicted class label Pᵢ $ Generate LIME explanations for prediction Pᵢ:
9: VRᵢ ← LIME(Pᵢ)
10:    Store LIME visualization VRᵢ
11: End For
12: Return: Predictions P₁, P₂, P₃, P₄ and their corresponding LIME visualizations VR₁, VR₂, VR₃, VR₄
```

## A. Problem Formulation

The CAN bus is a critical communication protocol in modern vehicles, enabling real-time data exchange between Electronic Control Units (ECUs). However, due to its lack of built-in security mechanisms, the CAN bus is highly susceptible to various forms of cyberattacks. This study aims to develop a lightweight and robust IDS capable of identifying malicious CAN messages using a benchmark dataset provided by the HCRL. The dataset contains different attributes for each: timestamp ($t$), CAN ID ($ID$), Data Length Code ($DLC$), data bytes ($\text{DATA}[0]\ldots\text{DATA}[7]$), and a flag ($Flag$) to classify between injected ($Flag = T$) and normal class samples ($Flag = R$). The injected (malicious) messages comprise four well-known attack types: Fuzzy Attack, RPM Spoofing Attack, Gear Spoofing Attack, and DoS Attack.

Let the set of CAN messages be defined as:

$$X = \{x_1, x_2, \ldots, x_n\}, \tag{1}$$

$$x_i = \{t_i, ID_i, DLC_i, \text{DATA}_i, F_i\}, \tag{2}$$

where each $x_i$ represents a single CAN message.

Given the labeled dataset $D = \{(x_1, y_1), (x_2, y_2), \ldots, (x_n, y_n)\}$, the goal is to accurately classify and assign a label $y_i$ to each message $x_i$, such that:

$$y_i = \begin{cases} 1 & \text{if } x_i \text{ is injected message - T,} \\ 0 & \text{if } x_i \text{ is benign message - R.} \end{cases} \tag{3}$$

## B. Dataset Acquisition and Preprocessing

This study utilizes a state-of-the-art HRCL dataset that is acquired from the Hyundai YF Sonata by connecting a Y-cable to the vehicle's OBD-II port located beneath the steering wheel. A Raspberry Pi 3 is then used to interface with the CAN bus, while a laptop computer is connected to the Raspberry Pi via WiFi for data acquisition [27].

The dataset contains four types of cyber-attacks targeting the CAN bus, namely: Fuzzy Attack, RPM Spoofing Attack, Gear Spoofing Attack, and DoS Attack. Each attack scenario file (in .csv format) contains both normal and malicious traffic. The details of the dataset, such as its description and number of samples/messages, are summarized in Table 2. Each CAN message in the dataset contains several key attributes: timestamp, CAN ID, Data Length Code (DLC), data payload (DATA[0]–DATA [7]), and a flag. The *timestamp* records the exact time in seconds when the message was captured. The *CAN ID*, represented in hexadecimal (e.g., 043f), identifies the type of message on the CAN bus. The *DLC* indicates the number of data bytes, ranging from 0 to 8. The *DATA[0]–DATA [7]* fields store the actual message content in byte format. The *flag* denotes the message type, where "T" indicates an injected (attack) message and "R" represents a regular (benign) message. The acquired dataset is preprocessed by converting hexadecimal values into their corresponding decimal format.

## C. Proposed Deep Learning Architecture

In this subsection, the detail of our proposed IDS is presented. The proposed IDS architecture consists of four blocks. Each block is optimized for efficient feature learning and classification. The input data after preprocessing is passed into the first block of the proposed model, as illustrated in Fig 2. The first block includes a convolutional layer activated by the GeLU function, followed by pooling layers to reduce spatial dimensions and enhance generalization. The second and

**Table 2. Description of Car Hacking Dataset.**

| Attack Name | Description | Total instances | No. of R instances | No. of T instances |
|---|---|---|---|---|
| Fuzzy Attack | Injects spoofed CAN messages with random IDs and data values at 0.5 ms intervals. | 3,838,860 | 3,347,013 | 491,847 |
| RPM Spoofing | Injects CAN messages with RPM-related IDs every 1 ms to spoof engine RPM data. | 4,621,702 | 3,966,805 | 654,897 |
| Gear Spoofing | Injects CAN messages with gear-related IDs every 1 ms to spoof gear shift data. | 4,443,142 | 3,845,890 | 597252 |
| DoS Attack | Floods the CAN bus with high-priority messages (e.g., ID '0x000') every 0.3 ms. | 3,665,771 | 3,078,250 | 587,521 |

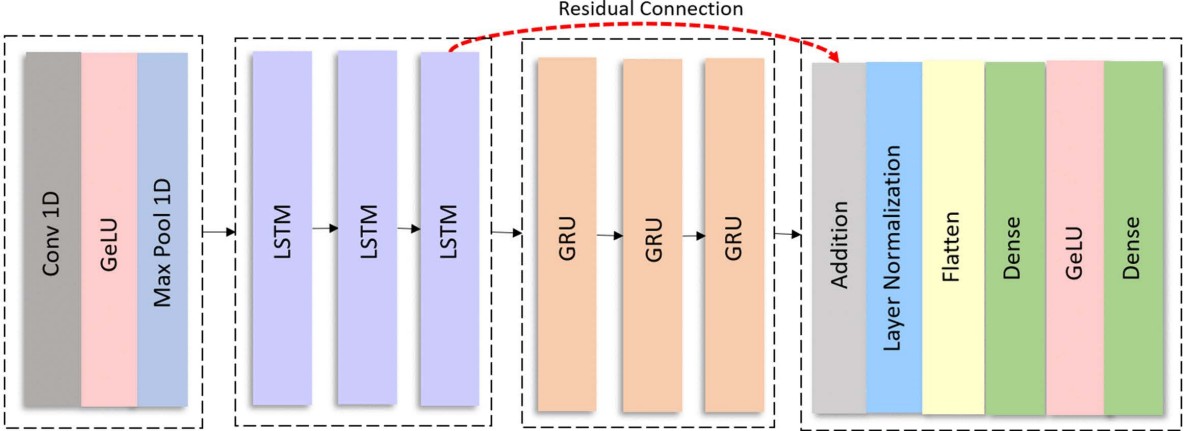

**Fig 2. Proposed DL and Residual Connection based IDS for AV Attack Detection.**

third blocks are composed of three LSTM layers and three GRU layers, respectively, to capture temporal dependencies in sequential CAN data. A residual connection is employed between the second and fourth blocks to improve gradient flow and training stability. The fourth block performs classification using addition, normalization, flattening, and two fully connected (dense) layers.

The proposed IDS takes an input of 10 features, including the Time Stamp, CAN ID, and DATA fields [0] – [7]. These input features encapsulate the temporal, identifier, and payload characteristics of each CAN message. The feature vector can be mathematically represented as: (4):

$$\mathbf{x} = \{x_1, x_2, \ldots, x_{10}\}, \quad x_i \in \mathbb{R} \tag{4}$$

where $x_1, x_2, \ldots, x_{10}$ represent each feature in the database.

The input feature vector is initially passed through a one-dimensional (1D) Convolutional layer to extract both spatial and temporal patterns from the sequential CAN bus data. This layer comprises 64 filters, each with a kernel size of 3, that slide over the input sequence to compute dot products between the filter weights and local input regions. The Convolutional layer employs the GeLU activation function [28], which introduces non-linearity by modulating input values based on the cumulative distribution function (CDF) of a standard normal distribution. This probabilistic activation function enhances model training by smoothing the transition between positive and negative activations to improve convergence and performance. The activation $h_{j1}$ of the $j$-th neuron in the Convolutional layer is computed as:

$$h_{j1} = \text{GeLU}\left(\sum_{i=1}^{k} W_{i1} x_{j+i-1} + b_1\right) \tag{5}$$

where $W_{i1}$ denotes the weight of the $i$-th element of the convolutional filter, $b_1$ is the bias term, $k = 3$ represents the kernel size, and $x_{j+i-1}$ corresponds to the input elements covered by the filter at position $j$. The resulting activation $h_{j1}$ is then forwarded to the next layers for further feature extraction.

Next, the features acquired from the Conv1D layer are passed to the Max Pool 1D layer that reduces the dimensionality and computational complexity by selecting the maximum value from a sliding window of size 2. By selecting a maximum value of 2, this operation helps in reducing the number of parameters while retaining the necessary features. The max pooling operation can be described using Eq. (6):

$$h_{j2} = \max(h_{2j1}, h_{2j+1}) \tag{6}$$

where $h_{j2}$ represents the activation after pooling.

Afterwards, the data is passed to an LSTM layer to extract spatial and temporal dependencies between the data. These layers are beneficial as they can retain long-range dependencies captured within the data. The LSTM cell contains four gates, i.e., forget gate, input gate, cell gate (also called candidate gate), and output gate. These gates are responsible for storing important information while discarding unnecessary information from the cell. The study deploys a stack of three LSTM layers. The output obtained from the final LSTM layer is then passed to GRU layers. This study uses three GRU layers with 32, 16, and 8 neurons, respectively. GRUs operate like LSTMs, but are simpler in terms of architecture and parameters. This makes these computationally efficient will still effective at capturing dependencies in the data.

The proposed IDS incorporates a residual connection between the final LSTM layer and the output of the last GRU layer that bypasses the intermediate layers. This connection (depicted as a dotted red arrow) helps in better gradient flow and enhances the learning process of the DL model. Residual connections are known for maintaining long-range dependencies in the data and aid the model during training by preserving important information and minimizing the vanishing gradients problem, or the risk of important information loss. Mathematically, the residual connection is defined as:

$$y_{l+n} = F(x_l) + x_l \tag{7}$$

Here, $x_l$ is the input from layer $l$, $F(x_l)$ represents the transformation applied across $n$ intermediate layers, and $y_{l+n}$ is the final output. The addition of $x_l$ to $F(x_l)$ allows the network to retain earlier information and simplifies learning by enabling identity mappings when needed. We applied a Normalization layer to standardize the output across the feature dimensions, which enhances the model's stability and convergence during the training phase. The normalization process is defined by Eq. (8):

$$\hat{x}^i = \frac{x^i - \mu}{\sqrt{\sigma^2 + \epsilon}} \tag{8}$$

Here, $\mu$ represents the mean, $\sigma^2$ is the variance, and $\epsilon$ is a small constant introduced to ensure numerical stability. It adjusts each feature to have a zero mean and unit variance, which improves the efficiency and reliability of the training process.

The normalized output is passed through a Flatten layer (containing 40 neurons) to reshape the data into a one-dimensional vector. Next, a Dense layer with 8 neurons is employed that applies a linear transformation, as defined in Eq. (9):

$$y = W \cdot x + b \tag{9}$$

Here, $W$ is the weight matrix, $x$ is the input, and $b$ is the bias. A GELU activation function is applied to introduce non-linearity.

The output layer (Dense) contains 2 neurons that correspond to the two classes, i.e., normal message and attack message. A softmax activation function is applied to transform the raw outputs into probabilities. The softmax function is defined in Eq. (10) as:

$$p(y = i \mid h) = \frac{e^{h_i}}{\sum_{j=1}^{2} e^{h_j}} \tag{10}$$

Here, $p(y = i \mid h)$ represents the probability of the input belonging to class $i$, and $h_i$ is the raw score (logit) for class $i$. The class with the highest probability is selected as the predicted output. Detailed configuration of the proposed IDS in terms of layer configuration and number of parameters is depicted in Table 3. The proposed IDS contains only 24,026 learnable parameters; hence, it is very lightweight. Moreover, the residual connection used in this study enhances the model's learning capability and helps manage gradient flow effectively.

## Results and discussion

This section presents a detailed discussion of the results obtained from the proposed IDS. Moreover, this section also includes a comprehensive comparison with existing systems and an evaluation using XAI techniques.

### A. Experimental Setup and Evaluation Metrics

All experiments reported in this section are conducted on the Kaggle platform with a P100 GPU. The car hacking dataset mentioned in Table 2 is used for training, validation, and testing of the proposed system. We used 70% of the dataset for training the models, whereas 20% dataset was used for validation purposes and 10% for testing. For statistical analysis on data splitting, we conducted experiments using three random seed points, i.e., 11, 42, and 99. Ablation analysis is also performed for optimization of the proposed architecture and hyperparameters.

 

**Table 3. Layer-wise specifications of the proposed IDS architecture.**

| Layer | Layer Configuration | Output | Number of Parameters |
|---|---|---|---|
| Input | Input Shape: 10 × 1 | 10 × 1 | – |
| Conv-1D | Filters: 64, Kernel Size: 3 | 10 × 64 | 256 |
| Activation | GELU Activation | 10 × 64 | – |
| MaxPooling-1D | Pool Size: 2 | 5 × 64 | – |
| LSTM 1 | Units: 32 | 5 × 32 | 12,416 |
| LSTM 2 | Units: 16 | 5 × 16 | 3,136 |
| LSTM 3 (Dropout) | Units: 8, Dropout: 0.2 | 5 × 8 | 800 |
| GRU 1 | Units: 32 | 5 × 32 | 4,032 |
| GRU 2 | Units: 16 | 5 × 16 | 2,400 |
| GRU 3 | Units: 8, Dropout: 0.2 | 5 × 8 | 624 |
| Add | Residual Connection | 5 × 8 | – |
| LayerNormalization | Normalization | 5 × 8 | 16 |
| Flatten | Flatten | 40 | – |
| Dense (Fully Connected) | Units: 8 | 8 | 328 |
| Activation | GELU | 8 | – |
| Dense (Classification – Softmax) | Units: 2 | 2 | 18 |
| **Total Number of Parameters** | | | **24,026** |

The proposed IDS is evaluated on accuracy, precision, recall, and F1-Score. Accuracy measures the performance of the proposed model in terms of correct predictions. A higher accuracy indicates the model's ability to distinguish between positive and negative classes. The accuracy can be calculated by using Eq. (11).

$$\text{Accuracy} = \frac{TP_{attack} + TN_{normal}}{TP_{attack} + TN_{normal} + FP_{attack} + FN_{normal}} \quad (11)$$

The True Negative Rate determines the ability of the model in the detection of benign or attack-free packets. A low False Negative Rate (FNR) shows that a few malicious packets were missed by the model, or there might have been no false detections by the model. The False Positive Rate (FPR), however, indicates that the model wrongly classified attack-free packets as malicious. A robust classification model is expected to have a low FPR.

However, accuracy alone is insufficient to evaluate automated systems; hence, we employed precision, recall, and the F1-score to assess the performance of the proposed IDS. Precision evaluates the effectiveness of a model in correctly identifying positive predictions by representing the proportion of true positives out of all predicted positives. Precision is measured as the ratio of $TP_{attack}$ predictions to the sum of true positive and $FP_{attack}$ predictions. The metric is calculated in Eq. (12).

$$\text{Precision} = \frac{TP_{attack}}{TP_{attack} + FP_{attack}} \quad (12)$$

Recall is a widely recognized performance metric used in classification tasks to evaluate a model's ability to correctly identify all positive cases in a dataset. It is also known as sensitivity or the true positive rate. It is calculated as the ratio of $TP_{attack}$ predictions to the sum of $TP_{attack}$ and $FN_{normal}$ predictions. Recall is calculated in Eq. (13).

$$\text{Recall} = \frac{TP_{attack}}{TP_{attack} + FN_{normal}} \quad (13)$$

F1-Score is a performance metric to combines both precision and recall into a single metric, thus achieving a balance between the two. The F1-Score is defined as a harmonic mean of precision and recall, with values ranging between 0 and 1, where 1 is the highest attainable score, denoting a perfect recall and precision. F1-score is calculated in Eq. (14).

$$F1 = 2 \cdot \frac{\text{Precision} \cdot \text{Recall}}{\text{Precision} + \text{Recall}} \tag{14}$$

## B. Performance of the Proposed Method

This subsection presents the results obtained from the proposed IDS. The IDS was trained and validated using a publicly available dataset comprising four well-known automotive cyberattack types: DoS, Fuzzy, RPM Spoofing, and Gear Spoofing attacks.

Fig 3 displays the confusion matrices for each class, which are used to assess the performance of the proposed trained models by showcasing their ability to differentiate between positive and negative class samples. The four quadrants of a confusion matrix correspond to true negatives (correctly predicted negatives), false positives (negatives misclassified as positives), false negatives (positives misclassified as negatives), and true positives (correctly predicted positives). It compares the actual class labels in the dataset with the model's predictions. The first confusion matrix depicts the performance of the proposed IDS on the DoS class, which shows that the model correctly identified 309,730 instances as true negatives and 58,647 instances as true positives. There are no false positives or false negatives, meaning that the model

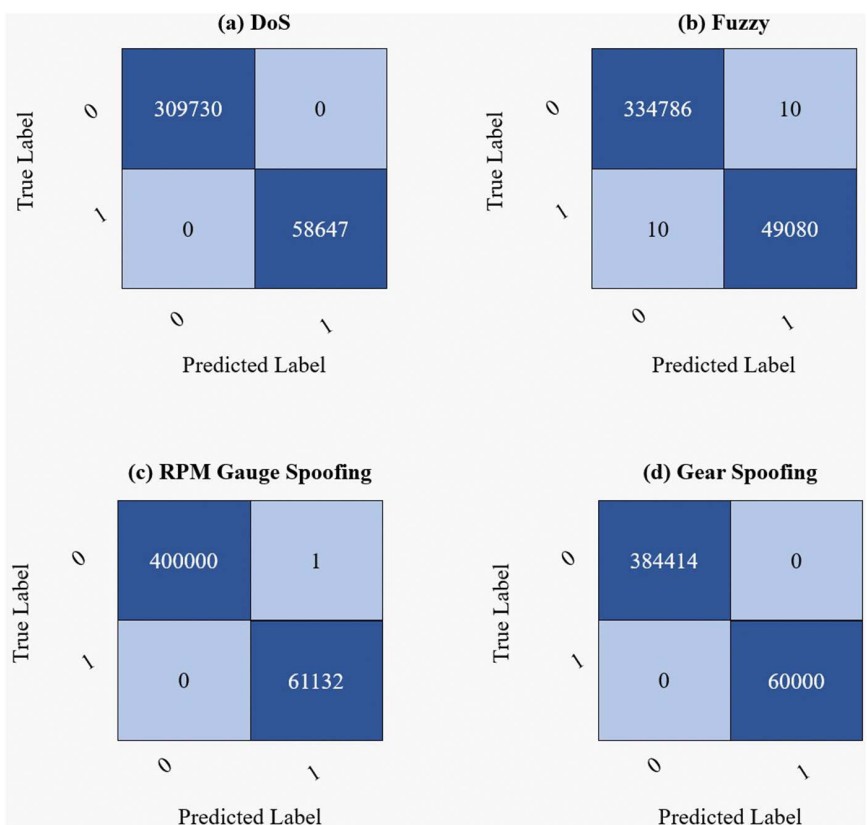

**Fig 3. Confusion matrix of the proposed IDS illustrating classification performance across different attack classes.**

did not make any mistakes in either predicting negative cases as positive or positive cases as negative, indicating a 100% accuracy. It is observed from the confusion matrix obtained from the Fuzzy class that the proposed model correctly identified 334,786 instances as true negatives and 49,080 instances as true positives. However, 10 false positive and 10 false negative cases were also observed, meaning the model misclassified 10 negative instances as positive and 10 positive instances as negative. Considering the large dataset size, these small errors are minimal, and the overall performance remains strong. The confusion matrix for the RPM Spoofing class shows that the model performs extremely well, as it correctly predicted 400,000 instances as true negatives and 61,132 instances as true positives. There is only one false positive, meaning that the model incorrectly classified one negative instance as positive. There are no false negatives in this matrix, indicating that all positive instances were correctly identified. Our proposed model is reliable due to its high detection accuracy. It is observed by the confusion matrix of the Gear Spoofing class that the proposed model correctly classified 384,414 instances as true negatives and 60,000 instances as true positives with no false positives or false negatives. These results are proof of the proposed framework's exceptional detection performance.

The performance of the proposed IDS can be seen in terms of precision, recall, F1-Score, and accuracy against DoS, Fuzzy, RPM Spoofing, and Gear Spoofing attacks in Fig 4. The observed results show that the proposed model obtained a perfect F1-score, Recall, and Precision of 1.0 in detecting DoS, RPM Spoofing, and Gear Spoofing attacks. It indicates that all the instances were correctly detected. The proposed model attained an F1-score, Precision, and Recall of 0.99 against the Fuzzy attack. It indicates that the proposed model made only a small number of errors. These results suggest that the model's performance remains strong and reliable, with only slight deviations from optimal performance. These

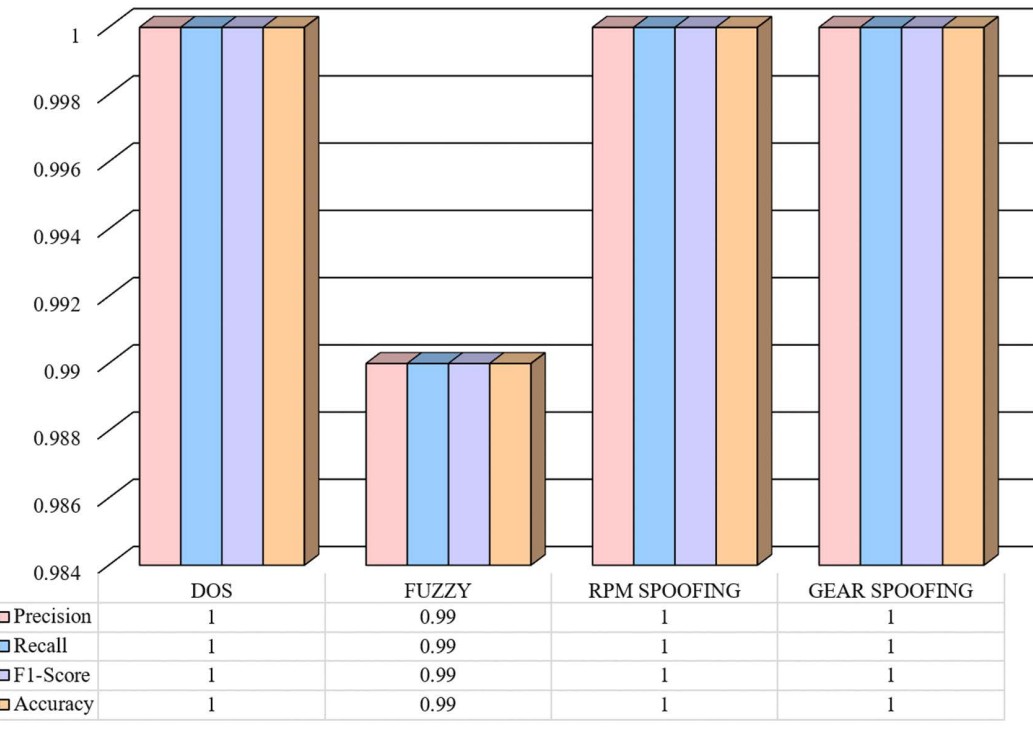

| | DOS | FUZZY | RPM SPOOFING | GEAR SPOOFING |
|---|---|---|---|---|
| Precision | 1 | 0.99 | 1 | 1 |
| Recall | 1 | 0.99 | 1 | 1 |
| F1-Score | 1 | 0.99 | 1 | 1 |
| Accuracy | 1 | 0.99 | 1 | 1 |

**Fig 4. Precision, Recall, F1-Score, and Accuracy Metrics for different attack classes obtained by the proposed IDS.**

observed results reflect that the proposed model performed exceptionally in detecting four attack types, and its performance was consistent (Fig 5).

Fig 6 shows four ROC (Receiver Operating Characteristic) curves corresponding to different attack classes: DoS, Fuzzy, Gear Spoofing, and RPM Spoofing. Each subplot contains two ROC curves, one for Class 0 (normal class) and one for Class 1 (attack class), and they evaluate the performance of a binary classification model for each type of attack. The ROC curve plots the true positive rate (sensitivity or recall) against the false positive rate at various threshold settings. A perfect classifier will have an ROC curve that touches the top-left corner of the plot, representing a high true positive rate with a low false positive rate. The area under the ROC curve (AUC) quantifies the overall ability of the model to discriminate between positive and negative classes, with an AUC of 1.0 indicating perfect classification. ROC curves for all the attack categories, i.e., DoS, Fuzzy, Gear Spoofing, and RPM Spoofing, show ROC perfect curves for Class 0 and Class 1, with an AUC of 1.0. The curves are represented as horizontal lines at the top of the plot, indicating that the model perfectly distinguishes between the two classes without making any false positive or false negative errors. The true positive rate reaches 1.0 immediately, and the false positive rate remains at 0. The results show the efficacy and robustness of the proposed IDS for the task of cyber attack detection in the CAN bus of AVs.

Table 4 presents the accuracy and loss values of the proposed IDS on four distinct attack types, evaluated during both training and testing phases. The proposed IDS obtained an accuracy of 0.99% on RPM Spoofing and Fuzzy attack classes during both training and evaluation phases. Whereas, the DoS and Gear Spoofing attained a perfect accuracy of 1.0, which indicates a flawless classification performance.

Table 5 shows the performance of the proposed IDS on the validation set. The system attained perfect precision, recall, and F1-scores across all classes, depicting a robust performance on unseen samples.

## C. Analysis of Data Splitting and Multi-Seed Reproducibility

We utilized multiple seed points to see the effect of dataset splitting on the performance of our proposed model. For each seed point, we created separate train, validation, and test splits for each attack file using the HCRL Car-Hacking dataset.

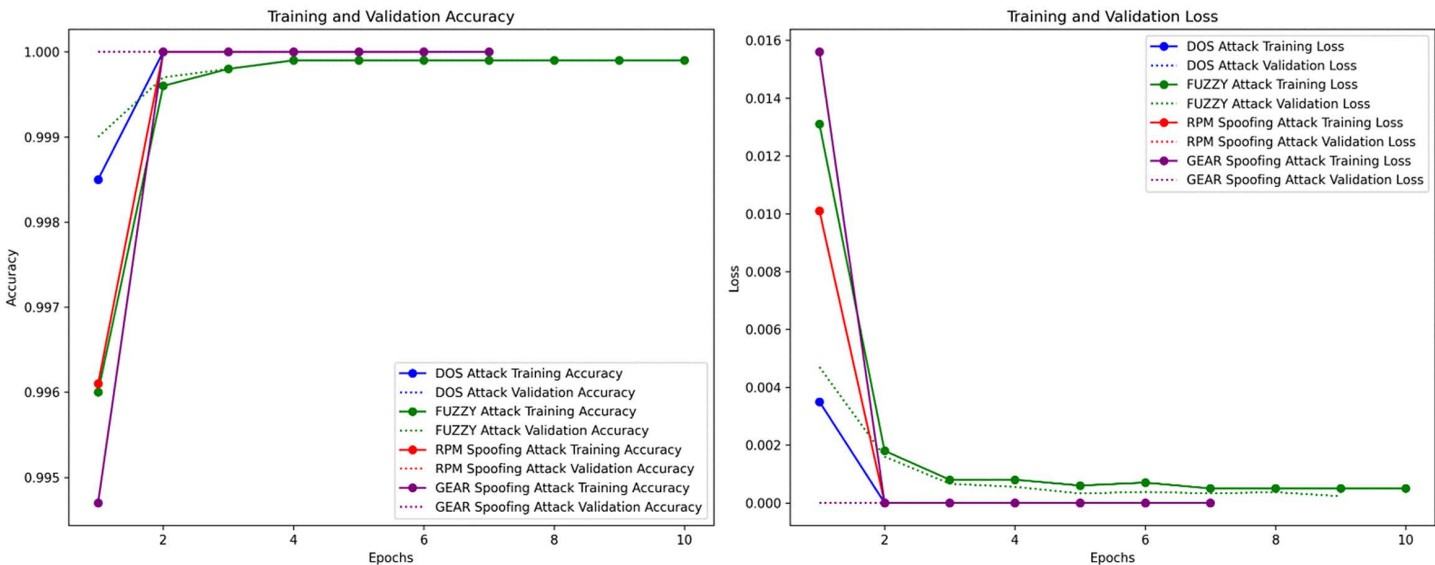

**Fig 5. Learning curves showing training and validation accuracy (left) and loss (right) of the proposed IDS.**

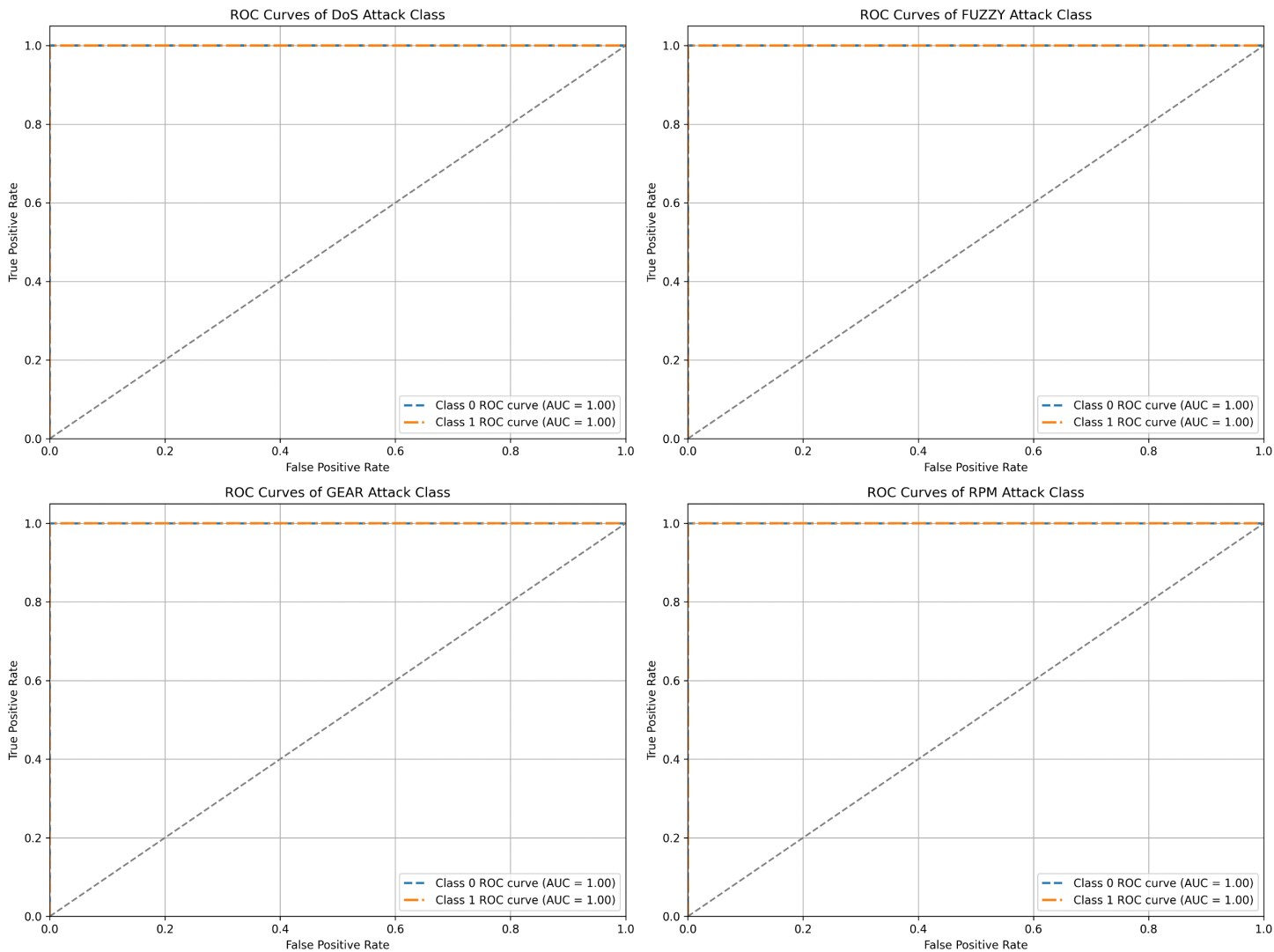

**Fig 6. ROC curves of the proposed IDS for various attack classes evaluated at different decision thresholds.**

**Table 4. Training and Testing Accuracy and Loss of the Proposed IDS Across Various Attack Classes.**

| Attack | Training | | Testing | |
|---|---|---|---|---|
| | Accuracy | Loss | Accuracy | Loss |
| RPM Spoofing | 0.99 | 0.0001 | 0.99 | 0.0004 |
| Fuzzy | 0.99 | 0.0001 | 0.99 | 0.0002 |
| DoS | 1.0 | 0 | 1.0 | 0 |
| Gear Spoofing | 1.0 | 0 | 1.0 | 0 |

The dataset provides separate CSV files for each attack type (DoS, Fuzzy, RPM Spoofing, Gear Spoofing). Each file was sorted by timestamp to keep the natural temporal order of CAN messages. Then, we applied a deterministic stratified splitting method that allocated 70% of the data to training, 20% to validation, and 10% to testing. Table 6 summarizes

**Table 5. Performance Metrics of the Proposed IDS on the Validation Dataset Across Various Attack Classes.**

| Attack | Precision | Recall | F1-Score | Accuracy |
|---|---|---|---|---|
| RPM Spoofing | 1.0 | 1.0 | 1.0 | 1.0 |
| Fuzzy | 1.0 | 1.0 | 1.0 | 0.99 |
| DoS | 1.0 | 1.0 | 1.0 | 1.0 |
| Gear Spoofing | 1.0 | 1.0 | 1.0 | 1.0 |

the dataset composition for each seed and split. The attack-to-normal ratio stays consistently between 10% and 20% across all seeds, confirming that the stratified, time-ordered splits maintain class balance and dataset size. We used three random seeds (11, 42, and 99) which resulted in twelve independent model trainings (four attack types multiplied by three seeds).

For each seed, we trained the models from scratch and tested on the corresponding test splits. Table 7 presents the aggregated test results as mean ± standard deviation including accuracy, precision, recall, F1-score, and ROC-AUC for each attack. The minimal variability between seeds shows that the proposed lightweight IDS delivers consistent performance regardless of random initialization or data split differences. We utilized an early stopping strategy during training so that the performance of the proposed model converges without overfitting. The confusion matrices in Fig 7, corresponding to the test splits in Table 6, remain strongly diagonal across seeds, and it shows that the model generalizes well to unseen CAN frames.

Although the dataset is synthetically generated under controlled in-vehicle conditions, the stable multi-seed results demonstrate that our model's performance reflects true separation between normal and attack traffic rather than data leakage artifacts.

## D. Robustness and Statistical Validation

To assess the stability and statistical reliability of the proposed IDS, additional robustness checks were performed beyond the multi-seed reproducibility analysis.

**Input perturbation (time and CAN jitter):** To mimic real-world sensor or timing noise, controlled Gaussian perturbations were added to the Timestamp and CAN ID fields, while individual DATA bytes were jittered by ±1 within the [0,255] range. Table 8 shows that deviations in accuracy and F1-score were negligible (<0.001), confirming the robustness of the IDS to input-level disturbances.

**Bootstrap confidence intervals:** To quantify statistical uncertainty on the test set, we computed non-parametric 95% bootstrap Confidence Intervals (CIs) (B = 1000 resamples) for F1 (computed from hard labels vs. hard predictions) and ROC-AUC (labels vs. soft probabilities) using seed 11. Results were extremely tight across all attacks as shown in Table 9. These intervals corroborate the stability of the observed near-perfect performance on the held-out test splits.

## E. Ablation Analysis and Parameter Optimization

In this work, we also conducted ablation analysis in order to assess the contribution of each component of our proposed architecture. The analysis involved four distinct architectural configurations, designated as four experiments, which can be observed in Table 10). In each experiment, we removed or added specific layers to assess their effect on detection performance. Each row in the table corresponds to a different experiment conducted on the Fuzzy Attack dataset, where certain layers or components of the model are either included (indicated by a checkmark) or omitted (indicated by a cross). The final column displays the resulting model accuracy for each configuration.

**Table 6.  Per-seed/per-split class counts and attack rates on the HCRL Car Hacking dataset.**

| Seed | Split | Attack | Total | Class 0 | Class 1 | Attack rate |
|---|---|---|---|---|---|---|
| 11 | Train | dos | 2,566,040 | 2,184,327 | 381,713 | 14.88% |
| | | fuzzy | 2,687,202 | 2,338,646 | 348,556 | 12.97% |
| | | rpm | 3,235,192 | 2,801,705 | 433,487 | 13.40% |
| | | gear | 3,110,199 | 2,713,523 | 396,676 | 12.75% |
| | Val | dos | 733,154 | 571,302 | 161,852 | 22.08% |
| | | fuzzy | 767,772 | 665,898 | 101,874 | 13.27% |
| | | rpm | 924,340 | 755,603 | 168,737 | 18.25% |
| | | gear | 888,628 | 736,220 | 152,408 | 17.15% |
| | Test | dos | 366,577 | 322,621 | 43,956 | 11.99% |
| | | fuzzy | 383,886 | 342,469 | 41,417 | 10.79% |
| | | rpm | 462,170 | 409,497 | 52,673 | 11.40% |
| | | gear | 444,315 | 396,147 | 48,168 | 10.84% |
| 42 | Train | dos | 2,566,040 | 2,174,101 | 391,939 | 15.27% |
| | | fuzzy | 2,687,199 | 2,330,002 | 357,197 | 13.29% |
| | | rpm | 3,235,192 | 2,786,715 | 448,477 | 13.86% |
| | | gear | 3,110,200 | 2,700,086 | 410,114 | 13.19% |
| | Val | dos | 733,154 | 568,123 | 165,031 | 22.51% |
| | | fuzzy | 767,772 | 668,057 | 99,715 | 12.99% |
| | | rpm | 924,340 | 755,953 | 168,387 | 18.22% |
| | | gear | 888,628 | 736,394 | 152,234 | 17.13% |
| | Test | dos | 366,577 | 336,026 | 30,551 | 8.33% |
| | | fuzzy | 383,889 | 348,954 | 34,935 | 9.10% |
| | | rpm | 462,170 | 424,137 | 38,033 | 8.23% |
| | | gear | 444,314 | 409,410 | 34,904 | 7.86% |
| 99 | Train | dos | 2,566,037 | 2,214,726 | 351,311 | 13.69% |
| | | fuzzy | 2,687,200 | 2,398,331 | 288,869 | 10.75% |
| | | rpm | 3,235,192 | 2,824,943 | 410,249 | 12.68% |
| | | gear | 3,110,198 | 2,732,727 | 377,471 | 12.14% |
| | Val | dos | 733,154 | 579,733 | 153,421 | 20.93% |
| | | fuzzy | 767,772 | 621,142 | 146,630 | 19.10% |
| | | rpm | 924,340 | 763,027 | 161,313 | 17.45% |
| | | gear | 888,628 | 746,883 | 141,745 | 15.95% |
| | Test | dos | 366,580 | 283,791 | 82,789 | 22.58% |
| | | fuzzy | 383,888 | 327,540 | 56,348 | 14.68% |
| | | rpm | 462,170 | 378,835 | 83,335 | 18.03% |
| | | gear | 444,316 | 366,280 | 78,036 | 17.56% |

In the first experiment, all components of the IDS architecture are included, including Conv1D, Max Pooling, three LSTM layers, three GRU layers, Residual Connection, and Dense layer. With all components present, the proposed IDS achieved an accuracy of 99.99%. In the second experiment, removal of the third LSTM and GRU layers resulted in 99.96% accuracy. In the third experiment, removal of Conv1D and Max Pooling layers resulted in 99.95% accuracy. In the fourth experiment, when all the components were included except for the residual connection, the model attained a 99.91% overall accuracy. The ablation experiments show that, when all the components were included, the proposed IDS attained the highest performance, thus proving the usability of the components in the accurate classification of cyber attacks in AVs.

**Table 7. Summary of model performance averaged across three random seeds (11, 42, 99) for each attack type in the HCRL Car-Hacking dataset. Values represent mean±standard deviation over independently trained models, showing consistent near-perfect detection performance and negligible inter-seed variance.**

| Attack | Accuracy | Precision | Recall | F1 | ROC-AUC |
|---|---|---|---|---|---|
| DoS | 1.00±0.0000 | 1.00±0.0000 | 1.00±0.0000 | 1.00±0.0000 | 1.00±0.0000 |
| Fuzzy | 1.00±0.0000 | 0.99±0.0001 | 1.00±0.0000 | 0.99±0.0000 | 1.00±0.0000 |
| RPM Spoofing | 1.00±0.0000 | 1.00±0.0000 | 1.00±0.0000 | 1.00±0.0000 | 1.00±0.0000 |
| Gear Spoofing | 1.00±0.0000 | 1.00±0.0000 | 1.00±0.0000 | 1.00±0.0000 | 1.00±0.0000 |

Along with the ablation study, extensive hyperparameter tuning is performed to find the optimal training parameters for the architecture. A random selection strategy was used to find and finalize the parameters. These parameters include learning rate, batch size, and number of layers, activation functions, etc. Table 11 depicts a detailed set of hyperparameters and their respective values that were evaluated during the optimization process for training a machine learning model. Bold text indicates the final values used for the model. During this parameter tuning process, we evaluated the performance of the proposed IDS by varying the values of epochs, batch size, learning rates, optimization function, no. of convolutional layers, neurons in LSTM/GRU layers, and the total number of LSTM, GRU, Convolutional layer, and Max Pooling layer. Furthermore, different optimization and activation functions were also tested to find the optimal set.

## F. Comparative Evaluation Against Existing Approaches

Table 12 provides a comparative analysis of the proposed IDS against existing systems in terms of F1-Score. The results demonstrate that the proposed IDS achieved an F1-score of 100% for DoS, Gear Spoofing, and RPM attacks, respectively, and 99.99% in the Fuzzy attacks database. In comparison, Amato et al. [7] obtained an F1-Score of 98.2%, 99.6%, and 100% on DoS, Fuzzy, Gear, and RPM Spoofing, respectively. Wang et al. [11] reported more than 99% F1-Score on different attacks, but their IDS did not cover Fuzzy attacks, whereas the proposed IDS detects four attacks with a remarkable performance. On the other hand, Sun et al. [17] reported very low F1-Scores on DoS and Fuzzy attack types. In comparison, the proposed IDS detects four well-known CAN bus attacks with remarkable performance compared to existing systems. The proposed IDS not only achieved phenomenal detection performance but is also very lightweight and robust due to a limited number of parameters.

A comparison of the proposed IDS with existing IDS systems in terms of parameters is shown in Table 13. The CAN protocol is built for real-time communication in environments with limited resources, it is unsuitable for handling computationally heavy or overly complex IDS. Given the constrained processing power and memory of automotive ECUs, an effective CAN-based IDS must be both lightweight and efficient, minimizing latency and resource consumption while still achieving high detection accuracy.

Computationally expensive IDSs alot are not suitable to be deployed in the AVs as they might consume alot of resources that might be required for other critical car functions, thus leading to accidents. Hence, the proposed IDS is very lightweight and robust to accurately and swiftly flag any malicious message. The results showcase the applicability of the proposed IDS in the evolving landscape of cyber threats in the automotive domain. Hence, the proposed IDS is reliable, robust, and lightweight for the task.

## G. Insights into Proposed Model Behavior through Explainable AI

LIME is utilized to interpret the predictions of the proposed deep approach for the detection of different types of attacks on the CAN bus. LIME generates local explanations by perturbing the input data and observing how the predictions change. This helps in identifying which features are most influential in making a particular prediction. By analyzing these perturbations, LIME can provide insights into the importance of each feature for a specific prediction, helping identify which

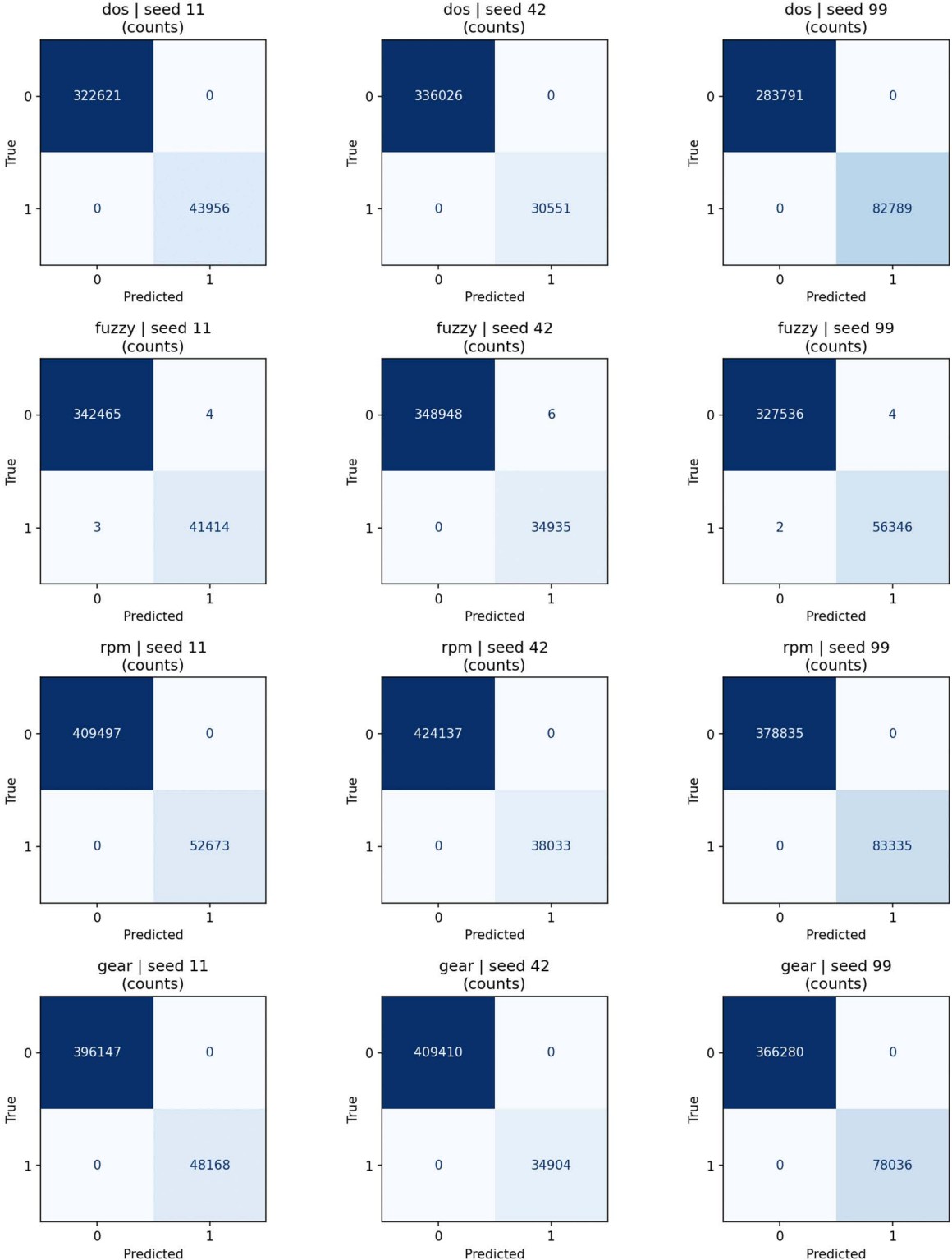

**Fig 7. Confusion matrices for test splits across multiple seeds and attack types.**

**Table 8. Robustness of the proposed IDS under test-time input perturbations (seed = 11). Gaussian noise was added to timestamps, CAN IDs were shifted by ±1, and DATA bytes were jittered by ±1 within [0,255].**

| Attack | Acc. (Base) | Acc. (Jitter) | ΔAcc. | F1 (Base) | F1 (Jitter) | ΔF1 |
|---|---|---|---|---|---|---|
| DoS | 1.0000 | 0.9999 | −0.00007 | 1.0000 | 0.9997 | −0.0003 |
| Fuzzy | 0.99997 | 0.99997 | +0.00000 | 0.99985 | 0.99987 | +0.00001 |
| RPM | 1.0000 | 1.0000 | 0.00000 | 1.0000 | 1.0000 | 0.00000 |
| Gear | 1.0000 | 0.99999 | −0.00001 | 1.0000 | 0.99998 | −0.00002 |

**Table 9. Bootstrap 95% confidence intervals (B = 1000) for F1 and ROC–AUC on the held-out test set (seed = 11).**

| Attack | F1 (mean [95% CI]) | ROC–AUC (mean [95% CI]) |
|---|---|---|
| DoS | 0.99998 [0.99994, 1.00000] | 1.00000 [1.00000, 1.00000] |
| Fuzzy | 0.99984 [0.99975, 0.99992] | 1.00000 [0.99999, 1.00000] |
| RPM | 1.00000 [1.00000, 1.00000] | 1.00000 [1.00000, 1.00000] |
| Gear | 1.00000 [1.00000, 1.00000] | 1.00000 [1.00000, 1.00000] |

**Table 10. Results of Ablation analysis of the proposed architecture for cyber attack detection in CAV.**

| Exp. | Conv1D | Max Pooling | LSTM 1 | LSTM 2 | LSTM 3 | GRU 1 | GRU 2 | GRU 3 | Res Connection | Dense | Accuracy |
|---|---|---|---|---|---|---|---|---|---|---|---|
| 1 | ✓ | ✓ | ✓ | ✓ | ✓ | ✓ | ✓ | ✓ | ✓ | ✓ | 99.99% |
| 2 | ✓ | ✓ | ✓ | ✓ | × | ✓ | ✓ | × | ✓ | ✓ | 99.96% |
| 3 | × | × | ✓ | ✓ | ✓ | ✓ | ✓ | ✓ | ✓ | ✓ | 99.95% |
| 4 | ✓ | ✓ | ✓ | ✓ | ✓ | ✓ | ✓ | ✓ | × | ✓ | 99.91% |

**Table 11. Optimized hyperparameters and their corresponding values used in the proposed IDS.**

| Parameters | Values |
|---|---|
| Epochs | 5, **10**, 20 |
| Batch Size | 64,128,**256** |
| Learning Rate | 0.01,**0.001**,0.0001 |
| Optimization function | SGD, **Adam**, RMSprop |
| No. of Convolution Layers | 0,**1**,2 |
| No. of Pooling layers | 0,**1**,2 |
| No. of LSTM and GRU layers | 1,2,**3** |
| LSTM/GRU Neurons | **8**, **16**, **32**, 64, 128, 256 |
| Activation Function | ReLU, **GeLU** |

aspects of the CAN bus data (e.g., message IDs, data length, data values, timestamp) contribute most to classifying a message as an attack or normal behavior.

The proposed model identifies anomalies in the CAN ID and DATA fields to make its predictions. LIME explanations for instances where attacks were detected (Class 1) are observed in Fig 8. The detailed explanation of the features driving the proposed model's decision to classify these instances as attacks can be seen clearly. LIME visualizations show features contributing to the attack classification highlighted in orange. For the DoS attack instance observed in Fig 8a, multiple DATA fields (like DATA [3], DATA [5], and others) show abnormal values, which suggest traffic congestion or high

**Table 12. F1-Score comparison of the proposed IDS with existing state-of-the-art methods on the Car Hacking dataset across four major attack types.**

| Technique | DoS | Fuzzy | Gear Spoofing | RPM Spoofing |
|---|---|---|---|---|
| Amato et al. [7] (2021) | 98.20% | 99.6% | 100% | 100% |
| Sun et al. [17] (2022) | 92.86% | 92.76% | – | – |
| Wang et al. [11] (2023) | 100% | – | 99.97% | 100% |
| Wickramasinghe et al. [23] (2023) | 99.67% | 99.51% | – | – |
| **Proposed Approach** | **100%** | **99.99%** | **100%** | **100%** |

**Table 13. Comparison of the proposed IDS with existing techniques in terms of model complexity (number of trainable parameters).**

| Reference | Technique | Parameters (in Millions) |
|---|---|---|
| Wang et al. [11] | DenseNet | 22.95 M |
| Yu et al. [12] | LSTM | 4.55 M |
| **Proposed IDS** | **LSTM, GRU, Residual Connection** | **0.0237 M** |

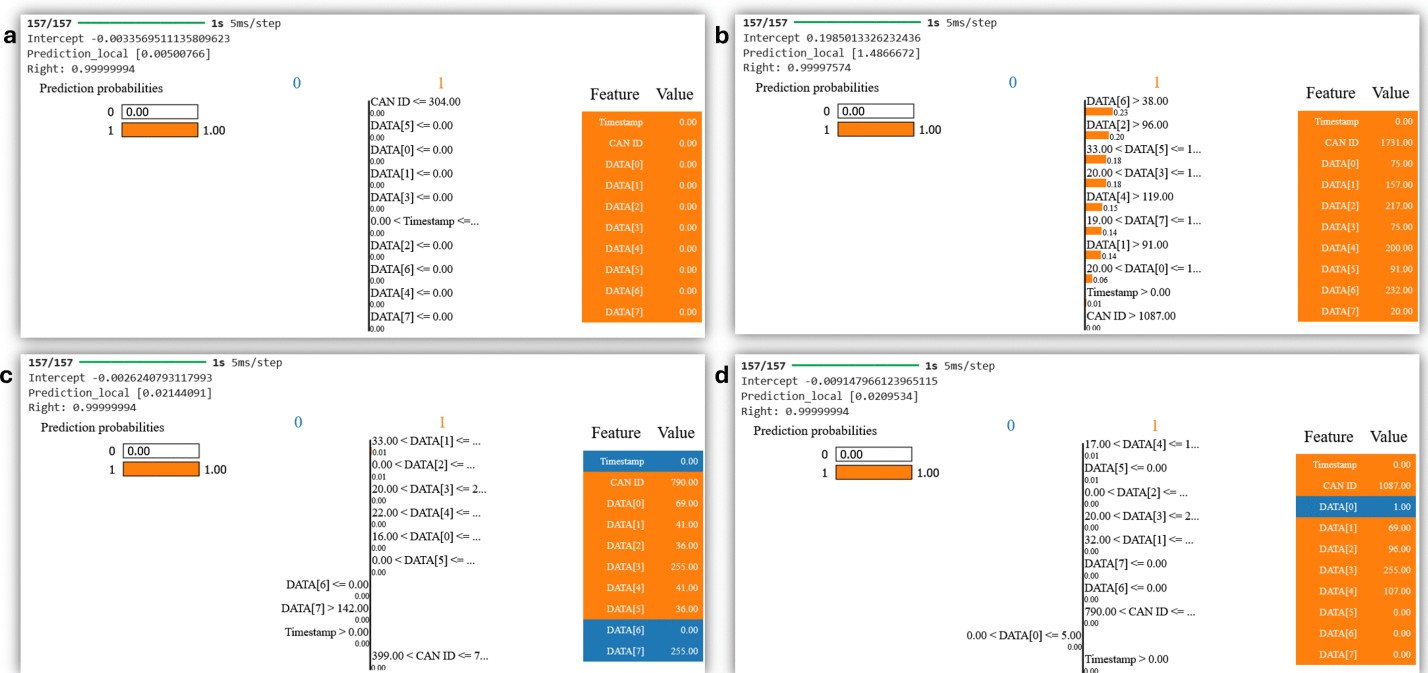

**Fig 8. LIME-based interpretability visualizations for attack instances classified as Class 1 by the proposed IDS.**

volume, typical characteristics of a DoS attack. In the case of Fuzzy attacks, certain DATA fields exhibit unusual, possibly random, values indicative of manipulated or corrupted data (Fig 8b). For the RMP attack, the CAN ID and other fields contribute to the detection, which indicates remote manipulation of the bus data and it can be observed in Fig 8c). Gear Spoofing attacks involve abnormal values in the DATA fields related to gear control, where the manipulation of these fields leads to the detection of the attack as observed in Fig 8d). LIME explanations show that certain fields, like CAN ID and

specific DATA fields, are key indicators of attack activity. This highlights the proposed model's capacity to capture subtle deviations in the CAN bus messages caused by different attack vectors.

It can be observed from these visualizations that the model relies on the stability of key data attributes such as the CAN ID and DATA field to classify instances as normal/attack-free.

LIME explanations for instances when no attacks were detected, i.e., Class 0, are observed in Fig 9. It shows four sub-plots, each representing a distinct form of attack: DoS, Fuzzy, RMP, and Gear Spoofing. It can be easily observed in each of these subplots of LIME explanation, how the proposed model predicts no attack (Class 0) using CAN bus data properties. The blue bars represent characteristics that help predict normal behavior. It is observed from these visualizations that our proposed model relies on the consistency of specific data (such as the CAN ID and DATA fields) to classify situations as non-attacked.

LIME's perturbation mechanism inherently enables 'what-if' sensitivity analysis by systematically modifying CAN message attributes and observing shifts in our proposed model predictions. Future work may extend this toward an interactive dashboard in order to allow practitioners to simulate and visualize system behavior under varying network or sensor conditions. It would support real-time diagnostic insights in AVs.

## H. Strengths and Limitations of the Proposed IDS

From the reported results, it is clear that the proposed IDS has key strengths. Our proposed IDS combines convolutional layers with LSTM and GRU units. We combined the hybrid models with the intention to improve the detection of intricate and sequential attack behaviors that are typical in AVs communications. Adding a residual connection between the recurrent layers was a deliberate choice to make training more stable and efficient. We can say that our system performed quite well based on the reported results as it achieves an accuracy of around 99.99% across four attack types, such as DoS, Fuzzy, RPM Spoofing, and Gear Spoofing. We also incorporated an explainability technique to give us insights into

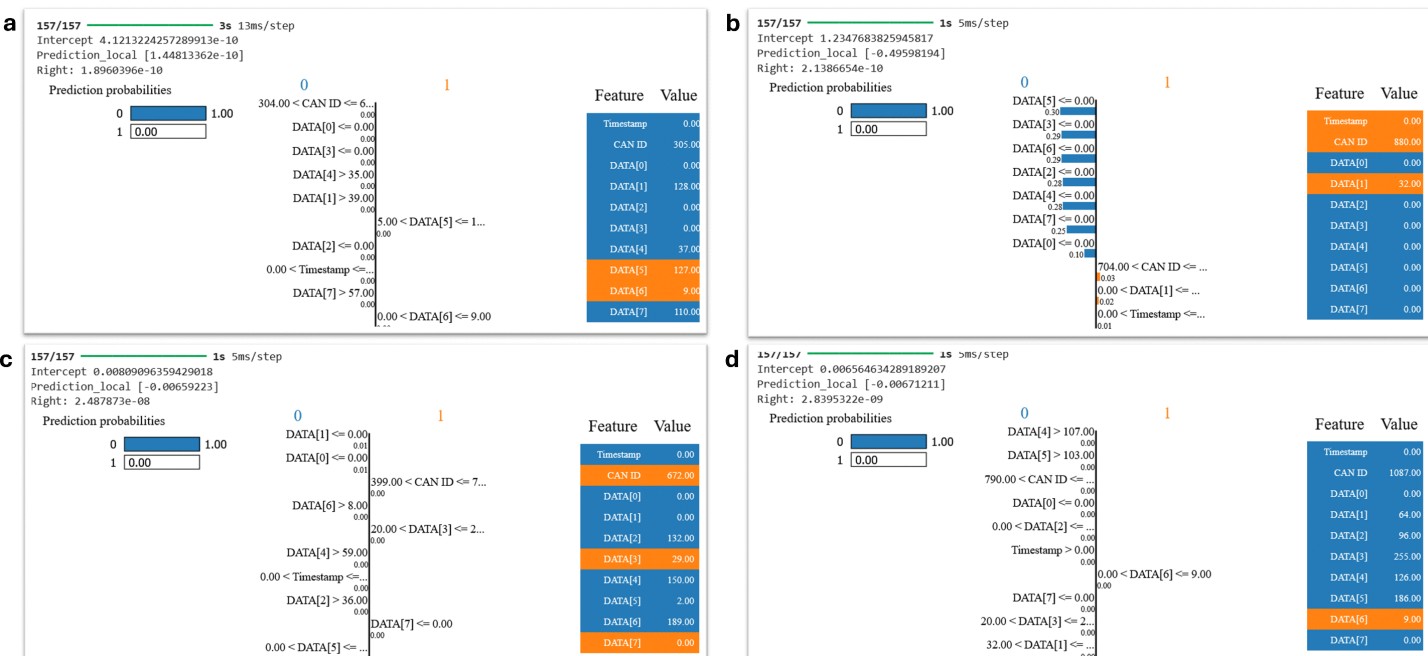

**Fig 9. LIME visual observations highlighting influential features for attack-free (Class 0) instances detected by the proposed IDS.**

how the model makes its decisions. It is important for trust and transparency in AVs. It is worth considering that despite our model's relatively small size (about 24K parameters or 94 KB), deploying it within CAV will introduce challenges related to inference latency, memory usage, and the realities of distributed embedded environments. We plan to investigate compression techniques, such as quantization and pruning, to further minimize resource demands. Exploring federated learning strategies could enable collaboration between ECUs without compromising raw CAN data privacy. These efforts aim to ensure IDS can be practically deployed on automotive microcontrollers and gateway units, which face strict requirements for both latency and safety.

## Conclusion and Future Work

Our proposed lightweight explainable IDS performs the detection of four types of cyber attacks in AVs with high detection accuracy. The proposed hybrid architecture is composed of convolutional layers with GeLU activation, followed by LSTM and GRU layers to effectively capture dependencies in the data. A residual connection is introduced between the LSTM and GRU blocks to improve gradient flow and training stability. We finalized this design after exhaustive hyperparameter tuning and ablation studies. The proposed IDS achieved 99.99% accuracy on a publicly available benchmark Car Hacking dataset containing well-known attacks, i.e., DoS, Fuzzy, and Spoofing. It is observed that the proposed IDS has significantly lower computational complexity compared to existing methods. The importance of integrating LSTM and GRU layers with residual connections can be observed from the results in improving the performance of the proposed model. Finally, the use of the XAI technique called LIME has increased the proposed model's transparency and trustworthiness in its decision-making. Future work may involve extending this framework to include emerging attack types to better adapt to the evolving and dynamic threat landscape in AVs.

## Author contributions

**Conceptualization:** Ayesha Siddiqa, Wazir Zada Khan, Atta ur Rehman Khan.

**Data curation:** Saad Alahmari, Saad Nasser Altamimi.

**Formal analysis:** Saad Alahmari.

**Funding acquisition:** Saad Alahmari, Saad Nasser Altamimi.

**Investigation:** Hareem Kibriya, Ayesha Siddiqa, Wazir Zada Khan.

**Methodology:** Hareem Kibriya, Ayesha Siddiqa, Wazir Zada Khan.

**Project administration:** Saad Alahmari.

**Resources:** Saad Alahmari, Saad Nasser Altamimi.

**Software:** Saad Alahmari, Saad Nasser Altamimi.

**Validation:** Hareem Kibriya, Ayesha Siddiqa, Wazir Zada Khan, Saad Nasser Altamimi, Atta ur Rehman Khan.

**Visualization:** Hareem Kibriya, Ayesha Siddiqa, Wazir Zada Khan.

**Writing – original draft:** Hareem Kibriya, Ayesha Siddiqa.

**Writing – review & editing:** Wazir Zada Khan, Atta ur Rehman Khan.

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
