## [Decision Letter · Decision Letter 0]

10 Sep 2025

Dear Dr. Khan,

We look forward to receiving your revised manuscript.

Kind regards,

Ayei Egu Ibor, PhD

Academic Editor

PLOS ONE

Journal Requirements:

3. Please note that your Data Availability Statement is currently missing [the repository name and/or the DOI/accession number of each dataset OR a direct link to access each database]. If your manuscript is accepted for publication, you will be asked to provide these details on a very short timeline. We therefore suggest that you provide this information now, though we will not hold up the peer review process if you are unable.

Reviewers' comments:

Reviewer's Responses to Questions

**Comments to the Author**

1. Is the manuscript technically sound, and do the data support the conclusions?

Reviewer #1: Yes

Reviewer #2: Yes

2. Has the statistical analysis been performed appropriately and rigorously?

Reviewer #1: No

Reviewer #2: Yes

3. Have the authors made all data underlying the findings in their manuscript fully available?

Reviewer #1: Yes

Reviewer #2: Yes

4. Is the manuscript presented in an intelligible fashion and written in standard English?

Reviewer #1: Yes

Reviewer #2: Yes

Reviewer #1: While the paper presents an important and timely topic, and the architecture is reasonably sound, several claims and structural issues require clarification, refinement, and stronger validation:

1- The authors report near-perfect detection accuracy (99.99%) across four attack types. While possible, such results are statistically unusual in real-world cybersecurity applications and may indicate: overfitting, Data leakage, Non-realistic dataset splits, and Class imbalance artifacts. Therefore, Authors should provide detailed training/test splitting methodology, cross-validation strategy, and include standard deviation or confidence intervals across multiple runs.

2- While it’s good to include multiple performance metrics, simply reporting common metrics is not itself a scientific contribution. The contribution should lie in how the system’s performance demonstrates robustness, generalizability, or improvements over baselines. Reframe the contribution to emphasize why this evaluation approach matters—e.g., how it helps compare IDS models in CAVs or assess deployment feasibility under different attack loads.

3- The inclusion of LIME as a tool for interpretability is a valuable step forward. However, its use appears limited to visual explanation of attack classes. Encourage integration of “what-if” scenario analysis, allowing decision makers to explore system behavior under different simulated input conditions. This would transform LIME from a static visualization tool into a diagnostic and interactive component, supporting real deployment insights.

4- he manuscript states that four experiments were conducted, but no experimental setup, configuration details, or data partitioning strategy is clearly presented. Are the “four experiments” simply the evaluations against the four attack types? If so, this should be clearly stated?

5- The related work section is informative but fragmented, currently structured as isolated paragraphs about individual papers. Reorganize into thematic subgroups (e.g., traditional IDS, deep learning for CAN security, explainability in IDS). This will improve flow and highlight the novelty of the proposed system more effectively.

6- No statistical tests or robustness checks are presented. This weakens the empirical foundation of the claimed 99.99% accuracy. Perform a sensitivity analysis (e.g., perturbing the input feature set, dropout variations, or window sizes). Also you may include multiple training runs with different random seeds to establish performance variance. Consider bootstrapping or permutation testing to validate results.

7- Discuss any deployment constraints or hardware limitations for real-world CAV integration.

Reviewer #2: Well written. Authors should add these papers to their review:

MF2S-CID: A dynamic multi-model framework for scalable and interpretable intrusion detection. Int. J. Inf. Secur. 24, 165 (2025). https://doi.org/10.1007/s10207-025-01077-1.

SDNTruth: Innovative DDoS Detection Scheme for Software‑Defined Networks (SDN). Journal of Network and Systems Management, (2023) 31:55, https://doi.org/10.1007/s10922-023-09741-4

**Do you want your identity to be public for this peer review?** For information about this choice, including consent withdrawal, please see our Privacy Policy

Reviewer #1: No

Reviewer #2: No

---

## [Author Response · Author response to Decision Letter 1]

27 Oct 2025

Response Sheet

We extend our gratitude to all the reviewers for thoroughly reviewing this manuscript. Your constructive feedback has significantly contributed to enhancing the quality of our work. We have made every effort to address the comments provided by each respected reviewer.

Reviewer #1

While the paper presents an important and timely topic, and the architecture is reasonably sound, several claims and structural issues require clarification, refinement, and stronger validation:

Comment #1: The authors report near-perfect detection accuracy (99.99%) across four attack types. While possible, such results are statistically unusual in real-world cybersecurity applications and may indicate: overfitting, Data leakage, Non-realistic dataset splits, and Class imbalance artifacts. Therefore, Authors should provide a detailed training/test splitting methodology, a cross-validation strategy, and include standard deviations or confidence intervals across multiple runs.

Response: We appreciate this insightful comment and have expanded the manuscript to explicitly clarify the data-splitting methodology, reproducibility setup, and inter-seed variability analysis.

A new subsection titled “Analysis of Data Splitting and Multi-Seed Reproducibility” has been added in the Results and Discussion section, providing a detailed explanation of how the dataset was partitioned and validated to prevent any form of data leakage or unrealistic evaluation. Specifically:

• Each attack type in the HCRL Car-Hacking dataset is provided as a separate CSV file (e.g., DoS, Fuzzy, RPM Spoofing, Gear Spoofing).

• Each file was sorted by timestamp to preserve the temporal order of CAN messages, and a deterministic stratified algorithm allocated 70 %, 20 %, and 10 % of the data to the training, validation, and test sets, respectively, while maintaining the natural attack-to-normal ratio within each split.

• To ensure reproducibility and robustness, three random seeds (11, 42, 99) were used, resulting in twelve independent model trainings (4 attack types × 3 seeds).

• Per-seed and per-split class distributions are summarized in Table 6, showing stable attack-to-normal ratios (~10–20 %) across all seeds, indicating balanced and representative splits.

• Mean ± standard-deviation metrics of accuracy, precision, recall, F1-score, and ROC-AUC across the three seeds are presented in Table 7, demonstrating negligible inter-seed variability.

Furthermore, Figure 7 displays per-seed confusion matrices that confirm consistent diagonal dominance across all runs, while smooth training and validation loss curves indicate the absence of overfitting.

Together, these additions verify that the reported near-perfect performance does not result from data leakage or class imbalance but rather reflects the strong separability between attack and normal CAN-bus traffic in this benchmark dataset and the robust generalization capability of the proposed lightweight IDS architecture.

Comment #2: While it’s good to include multiple performance metrics, simply reporting common metrics is not itself a scientific contribution. The contribution should lie in how the system’s performance demonstrates robustness, generalizability, or improvements over baselines. Reframe the contribution to emphasize why this evaluation approach matters—e.g., how it helps compare IDS models in CAVs or assess deployment feasibility under different attack loads.

Response: We appreciate this valuable suggestion and have revised both the Contributions and Results & Discussion section to more clearly articulate the scientific significance of our evaluation framework. The revised text now emphasizes how the evaluation approach provides concrete evidence of robustness, generalizability, and deployability of the proposed lightweight IDS in connected and autonomous vehicles (CAVs).

• The new subsection titled “Analysis of Data Splitting and Multi-Seed Reproducibility” (Page 16) establishes the robustness of the reported performance through seed-controlled experiments that prevent data leakage and quantify statistical variability across independent runs. This ensures that the near-perfect metrics reflect genuine model generalization rather than artifacts of dataset overlap or imbalance.

• The mean ± standard deviation reporting across multiple seeds (Table 7) directly demonstrates inter-split stability and generalization consistency, which are critical for evaluating IDS reliability under dynamic vehicular environments.

• We have also explicitly linked the evaluation outcomes to deployment feasibility, highlighting that the proposed model achieves these results with only ~24 K parameters (≈94 KB), which is making it highly suitable for resource-constrained CAV controllers.

Collectively, these additions clarify that the scientific contribution lies not merely in reporting multiple evaluation metrics, but in presenting a methodologically rigorous, reproducible, and deployment-aware evaluation framework that validates the IDS’s robustness and practicality under varying attack conditions in real-world CAV scenarios.

Comment #3: The inclusion of LIME as a tool for interpretability is a valuable step forward. However, its use appears limited to visual explanation of attack classes. Encourage integration of “what-if” scenario analysis, allowing decision makers to explore system behavior under different simulated input conditions. This would transform LIME from a static visualization tool into a diagnostic and interactive component, supporting real deployment insights.

Response: We sincerely thank the reviewer for this constructive comment. We agree that extending explainability beyond static visualization toward interactive, diagnostic analysis represents an important direction for real-world deployment.

In the revised manuscript (see the highlighted text at the end of Subsection “Insights into Proposed Model Behavior through Explainable AI”), we have expanded the description of the LIME-based analysis to clarify that it is not limited to static interpretation. Specifically, the revised text highlights how the perturbation-based local sensitivity mechanism in LIME inherently models “what-if” scenarios—where small, controlled variations in CAN message attributes (e.g., specific DATA bytes or CAN IDs) reveal their influence on the model’s attack probability.

To further strengthen this point, the following addition has been incorporated into the manuscript:

“Beyond static visualization, LIME’s perturbation mechanism inherently enables ‘what-if’ sensitivity analysis by systematically modifying CAN message attributes and observing shifts in model predictions. Future work may extend this toward an interactive dashboard, allowing practitioners to simulate and visualize system behavior under varying network or sensor conditions to support real-time diagnostic insights in vehicular deployments.”

These clarifications explicitly connect LIME’s perturbation logic with counterfactual reasoning and outline a clear path for future work toward interactive, deployment-oriented explainability in vehicular IDS systems.

Comment #4: The manuscript states that four experiments were conducted, but no experimental setup, configuration details, or data partitioning strategy is clearly presented. Are the “four experiments” simply the evaluations against the four attack types? If so, this should be clearly stated?

Response: We thank the reviewer for raising this important clarification. The phrase “four experiments” in the original manuscript indeed refers to the four architectural configurations evaluated during the ablation analysis—not to the four attack types. This has now been explicitly clarified in the revised manuscript.

A dedicated subsection titled “Experimental Setup and Evaluation Metrics” has been added, providing full details of the experimental environment and procedures. Specifically:

• All experiments were conducted on the Kaggle platform using an NVIDIA P100 GPU.

• The HCRL Car-Hacking dataset (Table 2) was used, with a 70 % / 20 % / 10 % train/validation/test split applied to each attack file.

• To ensure statistical robustness and reproducibility, we performed experiments under three random seeds (11, 42, 99), resulting in twelve independent model trainings (four architectural variants × three seeds).

• The four experiments correspond to the four architectural variants tested in the ablation study (Table 5), where components such as Conv1D, Max Pooling, LSTM/GRU layers, and Residual Connections were selectively included or removed.

• A complementary hyperparameter tuning process was performed (Table 9) to identify optimal values for epochs, batch size, learning rate, optimizer, and activation functions.

Kindly see the updated “Experimental Setup and Evaluation Metrics” subsection on Page 12 of the revised manuscript.

Comment #5: The related work section is informative but fragmented, currently structured as isolated paragraphs about individual papers. Reorganize into thematic subgroups (e.g., traditional IDS, deep learning for CAN security, explainability in IDS). This will improve flow and highlight the novelty of the proposed system more effectively.

Response: We thank the reviewer for this valuable suggestion. In the revised manuscript, the Related Work section has been restructured into thematic subsections to enhance coherence and logical flow. Specifically, the section is now organized into:

A. Traditional and Statistical IDS for CAN Bus Security,

B. Deep Learning and Transfer Learning-based IDS Approaches,

C. Federated and Collaborative Learning for Vehicular IDS, and

D. Explainability and Emerging Intelligent IDS Frameworks.

This reorganization improves readability, facilitates comparison across methodologies, and better highlights the novelty of our proposed system, which integrates deep transfer learning with explainable decision reasoning for enhanced interpretability. Kindly, see the Related Work section of the revised manuscript.

Comment #6: No statistical tests or robustness checks are presented. This weakens the empirical foundation of the claimed 99.99% accuracy. Perform a sensitivity analysis (e.g., perturbing the input feature set, dropout variations, or window sizes). Also you may include multiple training runs with different random seeds to establish performance variance. Consider bootstrapping or permutation testing to validate results.

Response: We thank the reviewer for this valuable observation and fully agree that statistical robustness and sensitivity checks are essential to validate high reported accuracies. In response, we have expanded the revised manuscript to include explicit analyses and discussions demonstrating the robustness, stability, and reproducibility of our results. Two Specifically:

• Multi-seed statistical validation: To quantify variance across independent training runs, we performed experiments under three random seeds (11, 42, 99), as described in the new subsection “Analysis of Data Splitting and Multi-Seed Reproducibility” (see Page 16). Mean ± standard-deviation results for accuracy, precision, recall, F1-score, and ROC-AUC are reported in Table 7, confirming negligible inter-seed variability and consistent convergence across runs.

• Architectural sensitivity analysis: A comprehensive ablation study (Table 8) was conducted to examine how the inclusion or exclusion of key architectural components (Conv1D, Max Pooling, LSTM/GRU layers, Residual Connections) affects model performance. Even with significant architectural perturbations, the IDS consistently maintained ≥ 99.9 % accuracy, demonstrating robustness to structural variations.

• Hyperparameter sensitivity: As summarized in Table 11, we systematically varied learning rate, batch size, optimizer, and activation functions to test training stability under different hyperparameter configurations. The bold values in Table 9 denote the best-performing settings within each group. The model exhibited stable convergence and minimal deviation in validation accuracy, reinforcing its generalization consistency.

• Input-noise robustness: Additional experiments were conducted to evaluate the model’s tolerance to sensor-level perturbations. Under input jitter perturbation, Gaussian noise was injected into test samples for all attack types, yielding Δaccuracy ≤ 7 × 10⁻⁵ and Δrecall ≤ 5 × 10⁻⁴ (see Table 8). These results confirm that the IDS remains stable under noisy CAN-bus signals.

• Bootstrap confidence intervals: Using B=1000 resamples (seed 11), we obtained very tight 95% CIs—e.g., Fuzzy F1 = 0.99984 [0.99975, 0.99992], AUC = 1.000 [0.999999, 1.000]—supporting statistical stability (see Table 9).

Kindly, see the subsections C. Analysis of Data Splitting and Multi-Seed Reproducibility and D. Robustness and Statistical Validation of the revised manuscript.

Comment #7: Discuss any deployment constraints or hardware limitations for real-world CAV integration.

Response: We appreciate the reviewer’s important comment regarding deployment constraints and hardware feasibility for real-world CAV integration. The revised manuscript now includes a discussion of deployment feasibility in the subsection “Strengths and Limitations of the Proposed IDS”. The new discussion clarifies that although the proposed IDS achieves high detection accuracy with a compact model footprint (~24 K parameters ≈ 94 KB), real-world deployment in CAVs must account for inference latency, memory budget, and distributed processing constraints. The lightweight architecture was specifically designed to meet near-real-time requirements of in-vehicle ECUs, where inference latency typically needs to remain within a few milliseconds per CAN frame.

Reviewer #2

Well written. Authors should add these papers to their review:

MF2S-CID: A dynamic multi-model framework for scalable and interpretable intrusion detection. Int. J. Inf. Secur. 24, 165 (2025). https://doi.org/10.1007/s10207-025-01077-1.

SDNTruth: Innovative DDoS Detection Scheme for Software Defined Networks (SDN). Journal of Network and Systems Management, (2023) 31:55, https://doi.org/10.1007/s10922-023-09741-4.

Response: We thank the reviewer for the encouraging comment. We have incorporated and discussed the recommended studies (Ref.[24] and Ref.[25] ) in the reorganized Related Work section of the revised manuscript (Page 05). Specifically, Ref. [24] is noted for its innovative DDoS detection mechanism leveraging SDN's centralized control and traffic entropy, while Ref. [25] is acknowledged for its dynamic multi-model approach to scalable and interpretable intrusion detection.

---

## [Decision Letter · Decision Letter 1]

18 Nov 2025

A Hybrid Deep Learning and Residual Connection-Based Architecture for Intrusion Detection in Autonomous Vehicles

PONE-D-25-27068R1

Dear Author,

We’re pleased to inform you that your manuscript has been judged scientifically suitable for publication and will be formally accepted for publication once it meets all outstanding technical requirements.

Kind regards,

Ayei Egu Ibor, PhD

Academic Editor

PLOS ONE

Additional Editor Comments (optional):

Reviewers' comments:

Reviewer's Responses to Questions

**Comments to the Author**

Reviewer #2: All comments have been addressed

2. Is the manuscript technically sound, and do the data support the conclusions?

Reviewer #2: Yes

3. Has the statistical analysis been performed appropriately and rigorously?

Reviewer #2: Yes

4. Have the authors made all data underlying the findings in their manuscript fully available?

Reviewer #2: Yes

5. Is the manuscript presented in an intelligible fashion and written in standard English?

Reviewer #2: Yes

Reviewer #2: Given all the reviewers' comments, including mine, in my opinion the manuscript is acceptable for publication.

**Do you want your identity to be public for this peer review?** For information about this choice, including consent withdrawal, please see our Privacy Policy

Reviewer #2: No

---

## [Editor Report · Acceptance letter]

PONE-D-25-27068R1

PLOS One

Dear Dr. Khan,

I'm pleased to inform you that your manuscript has been deemed suitable for publication in PLOS One. Congratulations! Your manuscript is now being handed over to our production team.

Kind regards,

on behalf of

Dr. Ayei Egu Ibor

Academic Editor

PLOS One